# Phytosterols Suppress Phagocytosis and Inhibit Inflammatory Mediators via ERK Pathway on LPS-Triggered Inflammatory Responses in RAW264.7 Macrophages and the Correlation with Their Structure

**DOI:** 10.3390/foods8110582

**Published:** 2019-11-16

**Authors:** Lanlan Yuan, Fan Zhang, Mingyue Shen, Shuo Jia, Jianhua Xie

**Affiliations:** State Key Laboratory of Food Science and Technology, Nanchang University, Nanchang 330047, China; 15270874205@163.com (L.Y.); Finefanzhang@163.com (F.Z.); jiashuo@email.ncu.edu.cn (S.J.); jhxie@ncu.edu.cn (J.X.)

**Keywords:** phytosterols, anti-inflammatory, structure–activity relationship, pro-inflammatory mediators, phosphorylated extracellular signal-regulated protein kinase, RAW264.7 macrophages

## Abstract

Phytosterols, found in many commonly consumed foods, exhibit a broad range of physiological activities including anti-inflammatory effects. In this study, the anti-inflammatory effects of ergosterol, β-sitosterol, stigmasterol, campesterol, and ergosterol acetate were investigated in lipopolysaccharide (LPS)-induced RAW264.7 macrophages. Results showed that all phytosterol compounds alleviated the inflammatory reaction in LPS-induced macrophage models; cell phagocytosis, nitric oxide (NO) production, release of tumor necrosis factor-α (TNF-α), and expression and activity of pro-inflammatory mediator cyclooxygenase-2 (COX-2), inducible nitric oxide synthase (iNOS), and phosphorylated extracellular signal-regulated protein kinase (p-ERK) were all inhibited. The anti-inflammatory activity of β-sitosterol was higher than stigmasterol and campesterol, which suggests that phytosterols without a double bond on C-22 and with ethyl on C-24 were more effective. However, inconsistent results were observed upon comparison of ergosterol and ergosterol acetate (hydroxy or ester group on C-3), which suggest that additional research is still needed to ascertain the contribution of structure to their anti-inflammatory effects.

## 1. Introduction

Inflammation is a natural response of the immune system to damage from physical, chemical, and pathogenic factors [1]. A general inflammatory response is a host’s beneficial self-protection response to external challenges or tissue damage, which eventually make the body return to normal. However, chronic inflammation is involved in pathologies such as arthritis, asthma, multiple sclerosis, inflammatory bowel disease, and atherosclerosis, which are harmful to the body [2,3,4]. Macrophages play an important role in inflammation [5]. They can produce and secrete a variety of bioactive substances such as reactive oxygen species (ROS), nitric oxide (NO), and tumor necrosis factor-α (TNF-α) to improve the inflammatory response. Lipopolysaccharide (LPS), consisting of lipid A, core sugars, and O-antigen, constitutes the major component of the outer membrane of Gram-negative bacteria, it can elicit a strong immune response, and is a potent innate immune stimulator [6,7]. LPS can promote the secretion of pro-inflammatory cytokines and the transcription of immune-related genes [8], as well as specifically identify Toll-like receptors on macrophages. Macrophages stimulated by LPS can produce a variety of cytokines. Excessive LPS can cause excessive release of cytokines and lead to inflammatory reaction of macrophages; thus, it can be used to construct an inflammatory model [9,10].

Currently, although nonsteroidal anti-inflammatory drugs such as aspirin and phenylbutazone are used in the treatment of inflammation induced by tissue damage, they were shown to have harmful side effects in clinical practice [8]. Exploration of anti-inflammatory active substance attracted worldwide attention owing to the rapid increase in inflammatory diseases. Phytosterols, found in many commonly consumed foods, especially fats and oils, are one of the active ingredients of edible vegetable oils [11], and they exhibit a broad range of physiological activities including anti-inflammatory effects [12]. At present, more than 40 kinds of phytosterol compounds were isolated and identified from plants, such as sitosterol, campesterol, and stigmasterol [13,14,15,16]. A large number of other phytosterol compounds were also found in fungi, such as ergosterol and ergosterol acetate. The anti-inflammatory activity of phytosterol compounds from plants was studied, and the anti-inflammatory sterols found in plants were identified in many studies [17,18,19,20]. Phytosterol compounds can reduce the expression of the pro-inflammatory mediators, cyclooxygenase-2 (COX-2) and nitric oxide synthase (iNOS), in LPS-stimulated RAW264.7 macrophages [21,22]. Phytosterols extracted from fungi were found to be mainly ergosterol and its derivatives, and it is reported that phytosterols are the anti-inflammatory compounds of mushrooms. A study of the phytosterols from edible mushroom *Innotus obliquus* indicated that phytosterols significantly inhibited the level of interleukin-1β (IL-1β), IL-6, and TNF-α in RAW246.7 macrophage cells [23]. Ma et al. isolated phytosterol compounds from sclerotia of *I. obliquus*, and they indicated that ergosterol and ergosterol peroxide could inhibit NO production and nuclear factor-κB (NF-κB) luciferase activity in RAW246.7 macrophage cells [24].

Studies on the signaling pathway of phytosterols’ anti-inflammatory activity mainly involved the NF-κB and mitogen-associated protein kinase (MAPK) pathway. As for the NF-κB pathway, TNF-α increases NF-κB through enhancing the inhibitor of NF-κB (IκB) kinase β (IKKβ) function, and the ubiquitination of inhibitor of NF-κB 3 (IκB3), leading to the suppression of NF-κB, causes inflammatory responses [25,26,27]. Researchers reported that phytosterols decreased the production of pro-inflammatory factor TNF-α in the inflammatory response of mice [28]. A previous study also indicated that β-sitosterol reduced the release of some pro-inflammatory cytokines and chemokines such as TNF-α, which is consistent with the results of our study [29]. In addition, it was found that β-sitosterol can inhibit NF-κB translocation to the nucleus, while the activation of protein tyrosine phosphatase (SHP-1) affects the anti-inflammatory effect on macrophages [29]. As for the MAPK pathway, studies showed that ergosterol and ergosterol peroxide inhibit the LPS-induced DNA-binding activity of NF-κB and CCAAT/enhancer binding protein b (C/EBPb), as well as the phosphorylation of p38, c-Jun amino-terminal kinase (JNK), and extracellular signal-regulated protein kinase (ERK) MAPKs [30]. Ergosterol peroxide downregulates the expression of low-density lipoprotein receptor (LDLR), which is regulated by C/EBP and 3-hydroxy-3-methyl-glutaryl-coenzyme A (HMG-CoA) reductase (HMGCR) in RAW264.7 cells, leading to anti-inflammatory activity [31].

Although the anti-inflammatory effects of phytosterols were investigated before, the role that their structure plays in modulating inflammatory responses is still obscure. A previous study of sterols from corbiculid bivalve clam *Villorita cyprinoides* suggested that the anti-inflammatory activity of sterols is inversely related to their steric bulk, electronic, and hydrophobic features [32,33]. Taking into account that the anti-inflammatory mechanisms and structure–activity relationship of phytosterols remain to be fully examined, more studies should be carried out. In this study, as shown in Figure 1, five phytosterol compounds, ergosterol, β-sitosterol, stigmasterol, campesterol, and ergosterol acetate, commonly found in vegetables, were selected to compare their anti-inflammatory activity. LPS-induced RAW264.7 (mouse macrophage cell line) cells were employed as the anti-inflammation model to test the anti-inflammatory activities of ergosterol, β-sitosterol, stigmasterol, campesterol, and ergosterol acetate. The cell proliferation, phagocytic activity, and the levels of NO and inflammatory mediators such as TNF-α secreted by induced RAW264.7 macrophages were taken as the inflammatory indexes. Moreover, to thoroughly explore the mechanism of their anti-inflammatory activities, we further investigated proteins involved in the inflammatory response, including COX-2, iNOS, ERK, and p-ERK (phosphorylated ERK), in order to clarify whether special functional groups of phytosterols contribute to their anti-inflammatory activities.

## 2. Materials and Methods

### 2.1. Reagents

Phytosterol compounds including ergosterol (purity: 98%), stigmasterol (purity: 97%), β-sitosterol (purity: 98%), campesterol (purity: 98%), and ergosterol acetate (purity: 98%) were purchased from Toronto Research Chemicals Inc. (Thornhill Research Inc., Toronto, ON, Canada). The RAW264.7 cell line was purchased from the Type Culture Collection of Chinese Academy of Sciences (Shanghai, China). Dulbecco’s modified Eagle’s medium (DMEM) and fetal bovine serum (FBS) were obtained from Hyclone (GE Healthcare, Los Angeles, CA, USA). Cell Counting Kit-8 (CCK-8) was purchased from Japanese Dojindo Laboratories (Tokyo, Japan). Phosphate-buffered saline (PBS) was obtained from Chinese fir in Jinqiao (Beijing, China). Lipopolysaccharide (LPS), fluorescein isothiocyanate (FITC)-dextran, and neutral red were purchased from Sigma Chemical Co. (Saint Louis, MO, USA). NO Test Kit was purchased from Beyotime Biotechnology (Haimen, China). A TNF-α enzyme-linked immunosorbent assay (ELISA) kit for mice was obtained from Wuhan Boster Biological Technology (Wuhan, China). COX-2 and iNOS activity assay kits were purchased from Nanjing Jiancheng Bioengineering Institute. Enhanced bicinchoninic acid (BCA) Protein Assay Kit, lysis buffer, phenylmethanesulfonyl fluoride (PMSF), sodium dodecyl sulfate polyacrylamide gel electrophoresis (SDS-PAGE) sample loading buffer (5×), and Electrochemiluminescence (ECL) Kit were purchased from Beyotime Biotechnology (Haimen, China). Bull serum albumin (BSA) was bought from Sigma-Aldrich (St. Louis, MO, USA). Rabbit anti-COX-2, iNOS, ERK, and p-ERK antibodies were purchased from Cell Signaling Technology (USA). The β-actin antibody, antibodies against rabbit immunoglobulin (Ig) horseradish peroxidase (HRP) and antibodies against mouse Ig-HRP were obtained from Chinese fir in Jinqiao (Beijing, China). All other chemicals were of analytical reagent grade and were purchased from Shanghai Chemicals and Reagents Co. (Shanghai, China).

### 2.2. Cell Culture and Preparation of Phytosterols

RAW264.7 cells were recovered from a liquid nitrogen tank, and then cultured in DMEM supplemented with 10% fetal bovine serum (FBS) and 1% penicillin/streptomycin at 37 °C in an atmosphere containing 5% CO_2_. Cells were used for study when they attained approximately 70–80% confluence.

All phytosterol compounds were dissolved in complete medium to obtain 200-μM stock solutions. These phytosterols solutions were then filtered through a 0.22-μm membrane filter and stored in brown bottles at 4 °C for subsequent experiments.

### 2.3. Cell Proliferation Assays

The effects of tested compounds on the proliferation of RAW264.7 cells were simultaneously determined using CCK-8 assay. Briefly, RAW264.7 cells were seeded onto 96-well plates at a density of 10^4^ cells/well and cultured in an incubator maintained at 37 °C with 5% CO_2_ for 4 h. Afterward, the culture medium was removed and the cells were treated with 200 μL of culture medium containing 1 μg/mL LPS (final concentration, model group), or 200 μL of culture medium containing 1 μg/mL LPS (final concentration) and phytosterols at different concentrations (25, 50, 100, and 200 μM, experimental groups). Meanwhile, 200 μL of culture medium without phytosterols and LPS was added as a control group. After incubation for 24 h, the culture medium containing 10% CCK-8 was added, and the model, control, and experimental groups were incubated at 37 °C for 1 h. The absorbance value was measured at 450 nm on a microplate reader (Thermo Scientific, Waltham, MA, USA). Cell proliferation was calculated using the following equation:(1)$$\mathrm{Cell}\;\mathrm{proliferation}\;(\%)=\frac{A_{2}}{A_{1}}\times 100$$
where *A*_1_ is the absorbance of the control group, and *A*_2_ is the absorbance of the experimental or model group.

### 2.4. Neutral Red Experiment

RAW264.7 cells were seeded in 96-well plates at a density of 1 × 10^5^ cells/well, while wells with equal volumes of culture medium were set as the blank group. After 4 h of incubation, the culture medium was removed and cells were treated as mentioned above (Section 2.3). After incubation for 24 h, the culture medium was removed, and 100 μL of neutral red solution (1 mg/mL) was added to each well, followed by incubation for 30 min. Then, the culture medium was removed, and cells were washed three times with phosphate-buffered saline (PBS) to remove the neutral red which was not phagocytized by RAW264.7 macrophages. The cells were lysed with 100 μL of cell lysate (ethanol–glacial acetic acid = 1:1, *v*/*v*) in a fume hood at room temperature for 2 h. The absorbance value at 540 nm was recorded by a microplate reader (Thermo Scientific, Waltham, MA, USA) [34]. Cell phagocytic rate was calculated using the following equation:(2)$$\mathrm{Cell}\;\mathrm{phagocytic}\;\mathrm{rate}\;(\%)=\frac{A_{b}-A_{0}}{A_{C}-A_{0}}\times 100$$
where *A_b_* is the value of the control, experimental, or model group, *A_c_* is the value of the model group, and *A*_0_ is the value of blank group.

### 2.5. Flow Cytometry

RAW264.7 cells were seeded in six-well plates at a density of 1 × 10^6^ cells/well. After 4 h of incubation, the culture medium was removed, and the cells were treated with 2 mL of culture medium containing 1 μg/mL LPS (final concentration, model group), or 2 mL of culture medium containing 1 μg/mL LPS (final concentration) and phytosterols at different concentrations (25, 50, 100, and 200 μM, experimental groups). Meanwhile, 2 mL of culture medium without phytosterols and LPS was added as the control group. After incubation for 24 h, the culture medium was removed, and the cells were washed two times with PBS; then, 100 μL of FITC-dextran (1 mg/mL) was added, and the cells were covered with tin foil paper to protect from light and incubated at 37 °C for 1 h. After incubation, 1 mL of PBS was added. Then, RAW264.7 cells were harvested and washed three times with PBS, before being resuspended in 500 μL of PBS buffer. The stained cells were then analyzed by a BD FASCalibur flow cytometer (Franklin Lakes, NJ, USA) [35,36].

### 2.6. Measurement of NO Production

For the determination of NO production, a Griess assay kit was used. RAW264.7 macrophage cells under exponential growth were seeded for experiments at 1 × 10^6^ cells/mL concentration in six-well culture plates. After 4 h of incubation, the cells were divided into control, experimental, and model groups, and were treated as mentioned above (Section 2.5). After treatment for 24 h, cell culture supernatants were collected, and the NO production was determined according to the instructions of the Griess assay kit [37].

### 2.7. Cytokine Assay

The cytokine TNF-α secreted in the culture medium was measured using the ELISA method. RAW264.7 cells under exponential growth were seeded at a concentration of 1 × 10^6^ cells/mL in six-well culture plates, treated as above (Section 2.5). After that, the culture medium was taken out and centrifuged at 5000× *g* for 5 min. The supernatant was collected for ELISA assay, following the manufacturer’s instructions (TNF-α ELISA Kit, Wuhan Boster Biological Technology, Wuhan, China).

### 2.8. Preparation of Whole-Cell Extracts

RAW264.7 macrophage cells were treated as mentioned above (Section 2.5). Then, they were washed with PBS and incubated for 5 min at 4 °C with 200 µL of lysis buffer (1% Triton X-100, 150 mmol/L sodium chloride (NaCl) in 20 mmol/L 2-amino-2-(hydroxymethyl)-1,3-propanediol (Tris) buffer containing leupeptin inhibitors and sodium phosphatase inhibitors; pH 7.5) added with 1 mM PMSF. Cells were centrifuged at 12,000× *g* for 10 min at 4 °C; then, the supernatant was stored at −80 °C for protein detection and immunoblot analysis. Protein concentrations were determined by the Enhanced BCA Protein Assay Kit.

### 2.9. COX-2 and iNOS Activity Assay

RAW264.7 macrophage cells were treated as mentioned above (Section 2.5.). Then, they were washed with PBS and incubated for 5 min at 4 °C with 200 µL of lysis buffer (1% Triton X-100, 150 mmol/L sodium chloride (NaCl) in 20 mmol/L 2-amino-2-(hydroxymethyl)-1,3-propanediol (Tris) buffer containing leupeptin inhibitors and sodium phosphatase inhibitors; pH 7.5) added with 1 mM PMSF. Then, the cell extracts were determined by COX-2 and iNOS activity assay kits.

### 2.10. Western Blot Analysis

Protein extracts were mixed with one-quarter volume of SDS-PAGE sample loading buffer (5×), and then were heated to 95 °C for 5 min. Proteins (30 μg) were separated by SDS polyacrylamide gel electrophoresis and were transferred electrophoretically onto an immobilon membrane (Millipore Corporation, Billerica, MA, USA) using a blotter (Bio-Rad, Richmond, CA, USA) at 80 V for 30 min and then 120 V for 60 min. The membrane was blocked with Tris-buffered saline containing 0.05% Tween-20 (TBS-T) and 3% BSA for 1 h at room temperature. After washing with TBS-T, the membrane was incubated for 1 h at room temperature in TBS-T containing 0.1% BSA, and then 1000-fold diluted anti-COX-2, iNOS, ERK, and p-ERK tag antibodies were added. For detection, the membrane was allowed to bind to 10,000-fold diluted antibodies against rabbit Ig-HRP in TBS-T containing 0.1% BSA for 1 h at room temperature. The protein expression was normalized by β-actin protein expression level; in other words, the gray value of β-actin bands was used as the internal standard to calculate the relative expression of target proteins. The protein expression was assessed using an ECL Plus Western Blotting Detection System (GE Healthcare) with a luminescent image analyzer (Bio-Rad).

### 2.11. Statistical Analysis

Data of each group were expressed as means ± standard error of the mean (SEM). SPSS16.0 statistical software was used for statistical analysis. The statistical significance of the results was evaluated by one-way analysis of variance (ANOVA) followed by the Duncan’s multiple range test. Origin 8 software was used for graphic drawing. Win MID 2.9 software was used to analyze the results of flow cytometry. Quantity One software was used to analyze the results of Western blot. All means were calculated from six independent experiments, and the difference was considered significant when *p* < 0.05.

## 3. Results

### 3.1. Cell Proliferation

To avoid cytotoxicity, the proliferation of RAW264.7 cells stimulated by LPS and treated with various concentrations of ergosterol, stigmasterol, β-sitosterol, campesterol and ergosterol acetate were detected by the CCK-8 method. The concentrations of phytosterols used in this study were based on our pre-experiment, in which the cell activity was tested at the concentrations of 12.5, 25, 50, 100, 200, and 400 μM, where the results showed that no anti-inflammatory effect was observed at the concentration of 12.5 μM and that the phytosterols could not be dissolved completely at the concentration of 400 μM; thus, concentrations of 25, 50, 100, and 200 μM were chosen for this study. In our study, the cell proliferation of the model group was significantly higher than that of the control group when compared together with the experimental groups of 25, 100, and 200 μM, but no significant difference between model group and control group was observed when compared together with the experimental group of 50 μM (*p* < 0.05). As shown in Figure 2, when the experimental group of 25 μM was compared together with the control and model group, there was no significant difference in survival rate between control and experimental group, and the same non-cytotoxic effects were also observed for the experimental groups of 50 and 100 μM (*p* < 0.05). In addition, the cell proliferation of ergosterol, ergosterol acetate, and β-sitosterol groups at 200 μM was still not significantly different from that of the control group, while cell proliferation of the corresponding campesterol and stigmasterol groups was lowered and enhanced, respectively. LPS (1 μg/mL) could promote the proliferation of RAW264.7 cells, and 25–200 μM ergosterol, stigmasterol, β-sitosterol, campesterol, and ergosterol acetate exhibited little or no cytotoxicity on RAW264.7 cells (*p* < 0.05). It was reported that LPS stimulates and induces RAW264.7 macrophages, which in a short period of time can cause fluctuations in macrophage cell proliferation. The RAW264.7 macrophages were treated with LPS for 24 h, and the cell proliferation was obtained during the theoretical fluctuation period, and these results were in accordance with previous publications [38,39]. It was suggested that 1 μg/mL LPS slightly enhanced the proliferation, and 25–200 μM phytosterol compounds hardly influenced the proliferation of RAW264.7 cells, which indicated that the concentration of LPS and phytosterol compounds used was appropriate in the present study.

### 3.2. Effects of Phytosterol Compounds on Phagocytic Activity in LPS-Stimulated RAW264.7 Macrophages: Neutral Red Method

Phagocytosis is a self-protection mechanism of somatic cells, and the detection of phagocytic activity of macrophages is a method reflecting the immune status of the body [35]. Effects of different phytosterol compounds on phagocytic activity in LPS-stimulated RAW264.7 macrophages are presented in Figure 3A. The inhibitory effects of these phytosterols on phagocytic activity exhibited dose dependence, where ergosterol, campesterol, and β-sitosterol inhibited phagocytosis of neutral red at the lowest concentration of 25 μM, while ergosterol acetate and stigmasterol acted at the concentrations of 50 and 100 μM, respectively. At levels of 25 μM, the phagocytosis of ergosterol, campesterol, and β-sitosterol groups was significantly inhibited, and no inhibitory effects were observed for ergosterol acetate and stigmasterol groups (*p* < 0.05). As for the experimental group of 50 μM, the inhibitory effects of ergosterol acetate, ergosterol, and β-sitosterol groups were at the same level, followed by the campesterol group, while the stigmasterol group showed almost no significant inhibitory effects (*p* < 0.05). Next, when treated with 100 μM phytosterol compounds, there was no significant difference in the inhibition of phagocytosis among ergosterol acetate, ergosterol, and campesterol groups, while weaker inhibitory effects could also be observed in the β-sitosterol and stigmasterol groups (*p* < 0.05). When treated with 200 μM phytosterol compounds, ergosterol acetate, ergosterol, campesterol, and β-sitosterol exhibited similar inhibitory effects on phagocytic activity, and they were more effective than stigmasterol (*p* < 0.05). Taking into consideration the minimum effective concentration of phytosterols and the inhibitory effects of phytosterols at the same level, it can be concluded that stigmasterol produced the weakest inhibitory effects on phagocytosis of neutral red, followed by ergosterol acetate. The inhibitory effects of β-sitosterol and campesterol were not significantly different and were stronger than those of ergosterol acetate but weaker than those of ergosterol.

In order to explore the structure–activity relationship of these phytosterol compounds, the anti-inflammatory activity of phytosterols with different structures were compared in pairs: ergosterol and ergosterol acetate (the only difference is on C-3, hydroxy or ester group); β-sitosterol and stigmasterol (with or without a double bond on C-22); β-sitosterol and campesterol (methyl or ethyl on C-24). According to these results, the inhibition of phagocytosis in the ergosterol group was higher than that in the ergosterol acetate group; in other words, phytosterol compounds with a hydroxyl on C-3 were more effective than those with an ester on C-3 in the inhibition of phagocytosis. The inhibition of phytosterols on phagocytosis of neutral red was in the order of β-sitosterol (without a double bond on C-22) > stigmasterol (with a double bond on C-22), which suggested that, in the inhibition of phagocytosis of neutral red, phytosterols without a double bond on C-22 were more effective than those with a double bond on C-22. Moreover, the inhibitory effects of β-sitosterol and campesterol were not significantly different, which suggested that their inhibitory effects were not correlated with the functional groups (methyl or ethyl) on the C-24 position of phytosterols.

### 3.3. Effects of Phytosterol Compounds on Phagocytosis Activity in LPS-Stimulated RAW264.7 Macrophages: FITC-Dextran Method

The phagocytic activity of RAW264.7 macrophages in the control, experimental, and model groups was further explored and confirmed by the FITC-dextran method. In Figure 3B, the results from flow cytometry analysis showed that, compared with the model group, these phytosterol compounds significantly inhibited the phagocytic activity of RAW264.7 cells stimulated by LPS. All the phytosterols inhibited the phagocytosis of macrophages at the lowest concentration of 25 μM, where the inhibitory effects of ergosterol acetate, ergosterol, and β-sitosterol on phagocytic activity exhibited dose dependence, while the inhibition of ergosterol acetate and ergosterol improved with the increase in their concentrations, and β-sitosterol showed an opposite manner. As for the experimental group of 25 μM, the phagocytosis was inhibited by all phytosterols, and the highest inhibition was observed for the β-sitosterol group, followed by ergosterol or campesterol groups. The lowest inhibition was observed for the ergosterol acetate or stigmasterol groups. No significant differences in the inhibition of phagocytosis between ergosterol acetate and stigmasterol groups or between ergosterol and campesterol groups were found (*p* < 0.05). When RAW264.7 cells were treated with 50 μM phytosterols, similar inhibitory effects were observed. β-Sitosterol exhibited the strongest inhibitory effects, followed by ergosterol or ergosterol acetate. The inhibitory effects of campesterol were stronger than those of stigmasterol but lower than those of ergosterol acetate or ergosterol (*p* < 0.05). At the concentration of 100 μM, the inhibitory effects of ergosterol acetate, ergosterol, and β-sitosterol were similar, which were higher than those of campesterol; the lowest inhibition was observed for the stigmasterol group (*p* < 0.05). Next, at the concentration of 200 μM, ergosterol was the most effective, followed by β-sitosterol, ergosterol acetate, or campesterol. The lowest inhibition was observed for the stigmasterol group (*p* < 0.05). In Figure 3B, the general trend could be observed that β-sitosterol exhibited the highest inhibition activity, followed by ergosterol; campesterol and ergosterol acetate were less effective than ergosterol, and stigmasterol exhibited the lowest inhibition activity.

The inhibition on phagocytic activity of phytosterols was in the order of β-sitosterol (ethyl on C-24) > campesterol (methyl on C-24), β-sitosterol (without a double bond on C-22) > stigmasterol (with a double bond on C-22), and ergosterol (hydroxyl on C-3) > ergosterol acetate (ester group on C-3). It can be speculated that phytosterol compounds with a hydroxyl on C-3 may be more effective in inhibiting phagocytosis than those with an ester on C-3; these results were consistent with those detected by the neutral red method. Furthermore, the comparison between the inhibitory effects of β-sitosterol and campesterol suggested that phytosterols without a double bond on C-22 were more effective than those with a double bond on C-22, and that phytosterols with an ethyl on C-24 were more effective than those with a methyl on C-24.

### 3.4. Effects of Phytosterol Compounds on NO Production in LPS-Stimulated RAW264.7 Macrophages

It is reported that NO is an inflammometer which can modulate cellular signaling involved in inflammation. In RAW264.7 macrophages, detection of the NO production induced by LPS is considered as a convenient and credible method for anti-inflammation screening [40,41]. The Griess assay was used as an effective and efficient method to quantitate NO production [42,43]. According to the instructions of the Griess assay kit, the standard curve was obtained by a series of standard NaNO_2_ solutions, and the correlation coefficient (*R*^2^) was above 0.999. Then, concentrations of NO in the experimental or control group were calibrated and compared with those in the model group (as 100%) (Figure 4). Results showed that the NO production induced by LPS in RAW264.7 macrophages was significantly inhibited by phytosterol compounds, and the minimum effective concentration was 100 μM for all tested phytosterols (*p* < 0.05). Also, their inhibitory effects on NO production exhibited dose dependence. When RAW264.7 cells were treated with phytosterols at the concentration of 100 μM, the NO production in experimental groups was reduced significantly, and no significant differences in the inhibitory activity were observed between ergosterol, ergosterol acetate, campesterol, and β-sitosterol groups. The stigmasterol group exhibited the lowest inhibitory activity in NO production (*p* < 0.05). As for the experimental group of 200 μM, campesterol, ergosterol acetate, and β-sitosterol had similar inhibition activity in NO production. The inhibitory activity of ergosterol and ergosterol acetate was stronger than that of stigmasterol but lower than that of β-sitosterol (*p* < 0.05). The general trend, as observed in Figure 4A, was that ergosterol acetate, ergosterol, β-sitosterol, and campesterol exhibited similar inhibition activity, while stigmasterol exhibited the lowest inhibition activity.

According to these results, neither a hydroxyl or ester on C-3, nor an ethyl or methyl on C-24 affected inhibition of the production of NO in RAW264.7 macrophages. Also, the inhibition on NO production was in the order of β-sitosterol (without a double bond on C-22) > stigmasterol (with a double bond on C-22), which suggested that phytosterols without a double bond on C-22 may be more effective in inhibiting NO production than those with a double bond on C-22.

### 3.5. Phytosterol Compounds Reduce the Inflammatory Reaction by Scavenging Cytokine TNF-α in LPS-Stimulated RAW264.7 Macrophages

The production of cytokines, such as TNF-α, occurs in response to the inflammatory reaction induced by LPS in RAW264.7 macrophages. Therefore, the suppression of phytosterols on TNF-α production in inflammatory models was tested to indicate their anti-inflammatory activity [44,45,46]. TNF-α was detected by an ELISA kit, and the standard curve was obtained by a series of standard TNF-α solutions with a correlation coefficient (*R*^2^) above 0.99. Then, concentrations of TNF-α in the experimental or control group were calibrated and compared with those in the model group (as 100%). In Figure 4B, results showed that the TNF-α production induced by LPS in RAW264.7 macrophages was significantly inhibited by phytosterols, and the minimum effective concentration was 25 μM for tested phytosterols except ergosterol acetate (200 μM). Also, their inhibitory effects on TNF-α production exhibited dose dependence. At the concentration of 25 μM, the inhibition effects of β-sitosterol and stigmasterol on TNF-α production were similar, which were higher than those of ergosterol or campesterol (*p* < 0.05). As for the experimental groups of 50 μM, β-sitosterol exhibited the strongest inhibitory effect, followed by stigmasterol. The lowest inhibition was observed for the ergosterol or campesterol groups (*p* < 0.05). When treated with 100 μM phytosterol compounds, the β-sitosterol or stigmasterol groups exhibited the strongest inhibitory effects, followed by the ergosterol or campesterol groups (*p* < 0.05). Next, at the concentration of 200 μM, the TNF-α production was inhibited in all experimental groups, where the inhibition effects of β-sitosterol and stigmasterol were similar, higher than those of ergosterol or campesterol. The lowest inhibition was observed for the ergosterol acetate group (*p <* 0.05). The general trend, as observed in Figure 4B, was that β-sitosterol exhibited the highest inhibition activity, followed by stigmasterol. The inhibitory effects of ergosterol and campesterol were not significantly different, and they were lower than those of stigmasterol but higher than those of ergosterol acetate.

The inhibition of phytosterols on TNF-α production was in the order of β-sitosterol (without a double bond on C-22) > stigmasterol (with a double bond on C-22), β-sitosterol (ethyl on C-24) > campesterol (methyl on C-24), and ergosterol (hydroxyl on C-3) > ergosterol acetate (ester on C-3), which suggested that, in the inhibition of TNF-α production, phytosterols with a hydroxyl on C-3 were more effective than those with an ester on C-3, phytosterols without a double bond on C-22 were more effective than those with a double bond on C-22, and phytosterols with an ethyl on C-24 were more effective than those with a methyl on C-24.

### 3.6. Inflammatory Mediators Regulated by Phytosterols

To elucidate the molecular mechanism via which phytosterols inhibited inflammation in RAW264.7 macrophages, the impact of phytosterols on COX-2 and iNOS expression was studied. According to the results of Western blot analysis, phytosterols effectively suppressed the expression of LPS-induced COX-2 and iNOS (Figure 5). Among these phytosterols, ergosterol acetate, campesterol, and stigmasterol inhibited the expression of LPS-induced COX-2 and iNOS in a dose-dependent manner (*p* < 0.05) (Figure 6).

As shown in Figure 6A, when treated with 25 μM phytosterols, ergosterol showed the strongest suppression of LPS-induced COX-2 expression, followed by β-sitosterol, while weaker effects were observed on ergosterol acetate and stigmasterol groups and campesterol did not affect COX-2 expression significantly (*p* < 0.05). At the concentration of 50 μM, the effect of ergosterol acetate was at the same level as ergosterol, while weaker inhibition was observed for campesterol, β-sitosterol, and stigmasterol groups (*p* < 0.05). At the concentration of 100 μM, ergosterol acetate showed the strongest inhibition on COX-2 expression, followed by ergosterol and stigmasterol, and the weakest effects were found for campesterol and β-sitosterol groups (*p* < 0.05). At the concentration of 200 μM, the suppression was in the order of ergosterol acetate, ergosterol, and β-sitosterol > stigmasterol > campesterol (*p* < 0.05). In general, the ergosterol group showed the strongest inhibition, while weaker effects were observed for ergosterol acetate and β-sitosterol groups, and the lowest suppression of COX-2 expression was observed for stigmasterol and campesterol groups. The inhibition activity of ergosterol on COX-2 expression was higher than that of ergosterol acetate, while that of β-sitosterol (without a double bond on C-22, ethyl on C-24) was higher than that of campesterol (without a double bond on C-22, methyl on C-24) and stigmasterol (with a double bond on C-22, ethyl on C-24); all these results suggested that ergosterol (hydroxyl on C-3) was more effective than ergosterol acetate (ester on C-3) in suppressing COX-2 expression, with an ethyl on C-24 may be more effective than a methyl on C-24, and the lack of a double bond on C-22 may be more effective than with the presence of a double bond on C-22.

As shown in Figure 6B, at the concentration of 25 μM, the inhibitory effects of β-sitosterol on LPS-induced iNOS expression were similar to those of stigmasterol and were stronger than those of the other three phytosterols (*p* < 0.05). At the concentration of 50 μM, stigmasterol was the most effective, followed by β-sitosterol, and the weakest inhibitory effects were observed for ergosterol acetate, ergosterol, and campesterol groups (*p* < 0.05). At the concentration of 100 μM, β-sitosterol and stigmasterol were the most effective, followed by ergosterol acetate and campesterol, and the weakest inhibitory effects were observed for the ergosterol group (*p* < 0.05). At the concentration of 200 μM, the inhibitory effects among campesterol, β-sitosterol, and stigmasterol were similar, and were stronger than those of ergosterol acetate, while the weakest inhibitory effects were observed for the ergosterol group. As for the relative expression of iNOS, the general trend was that the stigmasterol and β-sitosterol groups showed the strongest inhibition, while weaker effects were observed for the ergosterol acetate and campesterol groups, and the lowest suppression was observed for the ergosterol group. The inhibition activity of β-sitosterol (ethyl on C-24) on iNOS expression was higher than that of campesterol (methyl on C-24), whereas the inhibition activity of ergosterol (hydroxyl on C-3) was lower than that of ergosterol acetate (ester on C-3); all these results suggested that phytosterols with an ethyl on C-24 may be more effective in suppressing iNOS expression than those with a methyl on C-24, and those with an ester on C-3 may be more effective than those with a hydroxyl on C-3.

Furthermore, the COX-2 and iNOS activity of RAW264.7 macrophages in the control, experimental, and model groups was further explored and confirmed by enzymatic activity assay kits. As shown in Figure 7, the effect of phytosterols on the activities of inflammatory enzymes (COX-2, iNOS) was consistent with the results of Western blotting. All phytosterols inhibited COX-2 and iNOS activities; the inhibition capability of phytosterols on COX-2 activity was in the order of β-sitosterol (without a double bond on C-22) > stigmasterol (with a double bond on C-22), β-sitosterol (ethyl on C-24) > campesterol (methyl on C-24), and ergosterol (hydroxyl on C-3) > ergosterol acetate (ester on C-3), and that of iNOS activity was in the order of β-sitosterol (ethyl on C-24) > campesterol (methyl on C-24) and ergosterol (hydroxyl on C-3) < ergosterol acetate (ester on C-3).

### 3.7. Phosphorylation of Extracelluar Signal-Regulated Kinase (ERK) Mediated by Phytosterols

To further examine the functional importance of phytosterols on the phosphorylation of extracellular signal-regulated kinase (p-ERK), we investigated the expression of ERK and p-ERK in macrophages stimulated with LPS in the presence (different concentrations) or absence of phytosterols. As shown in Figure 5 and Figure 6C,D, the levels of p-ERK and p-ERK/ERK were both decreased by treatment with phytosterols (*p* < 0.05). As shown in Figure 6C, at the concentration of 25 μM, all phytosterols showed effects on suppressing LPS-induced p-ERK (*p* < 0.05). At the concentration of 50 μM, campesterol and β-sitosterol showed obvious inhibitory effects, while other phytosterols did not affect the expression of p-ERK. As for the experimental groups of 100 μM, ergosterol acetate and β-sitosterol produced the strongest inhibition of p-ERK, followed by campesterol, while ergosterol and stigmasterol still did not affect the expression of p-ERK (*p* < 0.05). As for the experimental groups of 200 μM, ergosterol acetate and β-sitosterol produced the strongest inhibition of p-ERK, followed by ergosterol, campesterol, and stigmasterol (*p* < 0.05). Above all, the general trend could be observed that β-sitosterol showed the strongest inhibition, followed by ergosterol acetate, while weaker effects were observed for the campesterol group, and the lowest suppression was observed for the ergosterol and stigmasterol groups. The inhibition activity of β-sitosterol (without a double bond on C-22, ethyl on C-24) on p-ERK was higher than that of campesterol (without a double bond on C-22, methyl on C-24) and stigmasterol (with a double bond on C-22, ethyl on C-24); moreover, the inhibition activity of ergosterol (hydroxyl on C-3) was lower than that of ergosterol acetate (ester on C-3). All these results suggested that phytosterols without a double bond on C-22 may be more effective in suppressing p-ERK than those with a double bond on C-22, while an ethyl on C-24 may be more effective than a methyl on C-24, and an ester on C-3 may be more effective than a hydroxyl on C-3 .

The p-ERK/ERK content is shown in Figure 6D; at the concentration of 25 μM, ergosterol acetate, β-sitosterol, and stigmasterol showed the strongest suppression on LPS-induced p-ERK/ERK content, followed by ergosterol and campesterol (*p* < 0.05). When treated with 50 μM phytosterols, the inhibition of p-ERK/ERK content was in the order of stigmasterol > β-sitosterol > ergosterol acetate > campesterol > ergosterol (*p* < 0.05). As for the experimental groups of 100 μM, ergosterol acetate and stigmasterol produced the strongest inhibition of p-ERK/ERK content followed by β-sitosterol, while weaker effects could be observed for the ergosterol group, and campesterol had the weakest inhibition on LPS-induced p-ERK (*p* < 0.05). As for the experimental groups of 200 μM, ergosterol acetate produced the strongest inhibition of p-ERK/ERK content, followed by β-sitosterol and stigmasterol, while weaker effects could be observed for the ergosterol group, and campesterol had the weakest inhibition on LPS-induced p-ERK (*p* < 0.05). Above all, the general trend could be observed that the ergosterol acetate, β-sitosterol, and stigmasterol groups showed the strongest inhibition, followed by the ergosterol group, while weaker effects were observed for the campesterol group. The inhibition activity of β-sitosterol (ethyl on C-24) on p-ERK/ERK content was higher than that of campesterol (methyl on C-24), whereas the inhibition activity of ergosterol (hydroxyl on C-3) was lower than that of ergosterol acetate (ester on C-3); all these results suggested that phytosterols with an ethyl on C-24 may be more effective in suppressing phosphorylation of ERK than those with a methyl on C-24, whereas an ester on C-3 may be more effective than a hydroxyl on C-3.

## 4. Discussion

The LPS-induced RAW264.7 macrophage model is usually employed as a model for studies on the inflammatory response and inflammatory mechanisms. Different inflammatory mediators are secreted by LPS-induced macrophages, such as TNF-α and NO [47]. The phagocytosis of RAW264.7 macrophages is important for the uptake of antigens. The inflammatory cytokine TNF-α is involved in the pathogenesis of septic shock, while NO production plays an important role in killing microbes, parasites, and tumor cells [48,49,50,51]. In this study, the inhibitory effects of phytosterols on phagocytosis, NO production, and release of TNF-α in RAW264.7 macrophages induced by LPS were investigated. As inflammatory response is related to the expression of pro-inflammatory mediators such as iNOS and COX-2 in macrophages, and the NO production is regulated by iNOS, COX-2, and TNF-α levels [17], the expression of iNOS and COX-2 was also investigated. Furthermore, changes in pro-inflammatory mediator expression indicated that the related signaling pathway of inflammatory response in LPS-induced macrophages may be activated; thus, the phosphorylation level of ERK protein, which plays an important role in the related signaling pathway, was analyzed by Western blotting to explore the mechanism of the phytosterols’ anti-inflammatory effects.

Results showed that the phagocytic activity was significantly inhibited by the tested phytosterol compounds, which demonstrated that the administration of phytosterol compounds may have a certain effect on the immune response caused by inflammation. To detect the phagocytosis of RAW264.7 cells, two methods were chosen, the FITC-dextran method and neutral red method, which were widely used in previous studies [51]. The results of these two methods were comparable, and the FITC-dextran method contributed more to the comparison of inhibitory effects on phagocytosis between β-sitosterol and campesterol, showing a more definitive result, i.e., the inhibitory effects of β-sitosterol were stronger than those of campesterol. Similarly, the NO and TNF-α production in LPS-induced RAW264.7 cells treated with ergosterol, β-sitosterol, stigmasterol, campesterol, and ergosterol acetate was significantly reduced (*p* < 0.05), which means that the inflammatory response stimulated by LPS was recovered by these phytosterols. In other words, phytosterol compounds could relieve the inflammatory reaction induced by LPS.

There are two types of cyclooxygenase (COX-1 and COX-2) in the body; COX-1 is commonly expressed in tissues, while COX-2 is only expressed after stimulation, which is often used as an indicator of inflammation as it also interacts with other inflammation-related factors so as to affect the inflammatory process [52]. Meanwhile, when cells are stimulated by LPS or inflammatory cytokines, iNOS can be induced to catalyze the production of large amounts of NO [52]. In our study, results showed that phytosterols could downregulate the expression and activity of COX-2 and iNOS, and NO production, as well as iNOS expression and activity, was suppressed by phytosterols in LPS-stimulated macrophages, indicating that phytosterols may inhibit the production of NO by downregulating the expression of iNOS in LPS-induced RAW264.7 cells.

According to previous studies, we speculated that the anti-inflammatory activity of phytosterol compounds tested in this study are related to the MAPK pathway in macrophages [46]. MAPK belongs to the protein kinase family, including extracellular signal-regulated protein kinase (ERK), c-Jun N-terminal kinase (JNK), and p38 MAPK, which can transmit a variety of extracellular signals to intracellular cellular locations, and which participate in the expression of a variety of inflammatory factors. MAPK is regulated by the phosphorylation system, which phosphorylates specific serine and threonine residues of the target protein substrate to activate the expression of corresponding downstream proteins, thereby regulating the release of a variety of inflammatory factors. In addition, phosphorylation of MAPK protein can lead to the activation of the NF-κB signaling pathway and the expression of iNOS [53]. Our results found that phytosterols could inhibit the phosphorylation of ERK, suggesting that their anti-inflammatory effects may be related to the ERK signaling pathway, which could reduce the LPS-induced COX-2, iNOS, and TNF-α expression, and finally lead to the reduction of NO.

To the authors’ best knowledge, few previous studies were reported on the structure–anti-inflammatory activity relationship of phytosterol compounds. In this study, the anti-inflammatory activity of phytosterols with different structure were compared in pairs: ergosterol and ergosterol acetate (the only difference on C-3, hydroxy or ester group); β-sitosterol and stigmasterol (with or without a double bond on C-22); β-sitosterol and campesterol (methyl or ethyl on C-24). As shown in Table 1, our study found that ergosterol is more effective than ergosterol acetate in terms of certain anti-inflammatory activity, including phagocytosis, TNF-α production, and COX-2 expression and activity; however, ergosterol acetate was more effective in terms of iNOS/p-ERK level and p-ERK/ERK content. Thus, a deeper investigation is needed to identify the contribution of these functional groups to their anti-inflammatory effects. The inhibitory effects of β-sitosterol (without a double bond on C-22) on phagocytosis, NO/TNF-α production, and COX-2 and p-ERK expression were consistent and were all stronger than those of stigmasterol (with a double bond on C-22). Furthermore, the inhibitory effects of β-sitosterol on phagocytosis (FITC-dextran method), TNF-α production, COX-2/iNOS/p-ERK expression, and p-ERK/ERK level were consistent and were all stronger than those of campesterol. Overall, our results indicated that phytosterol compounds without a double bond on C-22 and with an ethyl on C-24 may be more effective for anti-inflammation. However, inconsistent results were also observed for the effects of phytosterol compounds with a hydroxy or ester group on C-3 in this study; there are other factors such as steric bulk, electronic, or hydrophobic features which may affect the anti-inflammatory activity of phytosterols [32,33]. Thus, deeper investigations of the relationship between the structure of phytosterols and their anti-inflammatory activity are still needed in the future, and this study provides a reference for further research on the structure–activity relationship of phytosterols.

## 5. Conclusions

In this study, the anti-inflammatory effects of phytosterols with different structure in RAW264.7 macrophages induced by LPS were investigated, and the results showed that inflammation responses were largely mediated via a reduction of phagocytic capacity and NO/TNF-α production, inhibiting the expression and activity of pro-inflammatory mediators COX-2, iNOS, and p-ERK. In addition, the anti-inflammatory activity of phytosterols with different structure were compared in pairs: ergosterol and ergosterol acetate (the only difference on C-3, hydroxy or ester group); β-sitosterol and stigmasterol (with or without a double bond on C-22); β-sitosterol and campesterol (methyl or ethyl on C-24). It can be deduced that the anti-inflammatory activity of β-sitosterol (without a double bond on C-22) was higher than that of stigmasterol (with a double bond on C-22), suggesting that the anti-inflammatory activity is correlated with the functional group (with or without a double bond) on the C-22 position of phytosterols. Similarly, β-sitosterol was more effective than campesterol, suggesting that the anti-inflammatory activity is correlated with the functional group (methyl or ethyl) on C-24 of phytosterols. However, comparison of the anti-inflammatory activity of ergosterol and ergosterol acetate was controversial in this study, suggesting that the functional group (hydroxy or ester group) on C-3 may play a role in the anti-inflammatory activity of phytosterols, but additional research is still needed to ascertain the contribution of structure to the anti-inflammatory effects. This study provides a reference for further research on the structure–activity relationship of phytosterols, indicating that ergosterol, β-sitosterol, stigmasterol, campesterol, and ergosterol acetate have potential for developing anti-inflammatory drugs or functional food additives.

## Figures and Tables

**Figure 1 foods-08-00582-f001:**
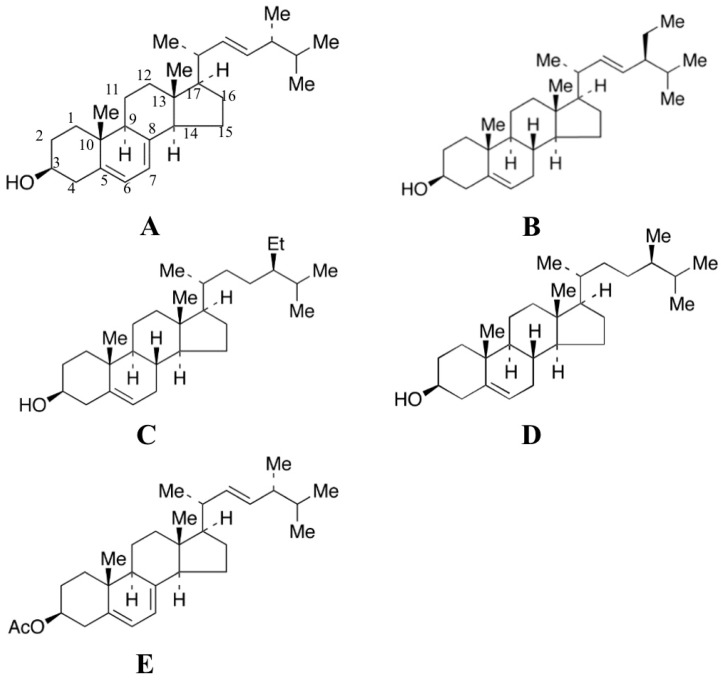
The structure of tested phytosterol compounds: (**A**) ergosterol; (**B**) stigmasterol; (**C**) β-sitosterol; (**D**) campesterol; (**E**) ergosterol acetate.

**Figure 2 foods-08-00582-f002:**
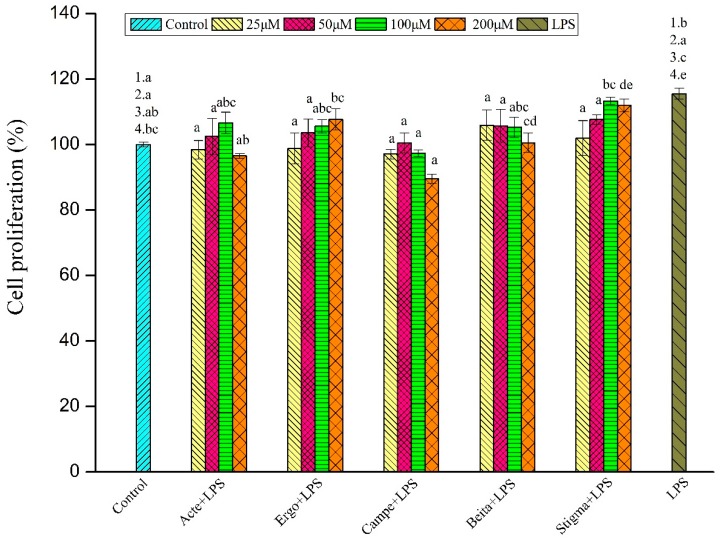
Effects of phytosterols on the proliferation of lipopolysaccharide (LPS)-stimulated RAW264.7 macrophages (*n* = 6). Acte (ergosterol acetate); Ergo (ergosterol); Campe (campesterol); Beita (β-sitosterol); Stigma (stigmasterol). The data are expressed as means ± standard error of the mean (SEM); different superscript letters for columns at the same concentration denote a significant difference (*p* < 0.05); the numbers describe that the control/model group was compared with the experimental group at concentrations of 25 (1), 50 (2), 100 (3), and 200 (4) μM.

**Figure 3 foods-08-00582-f003:**
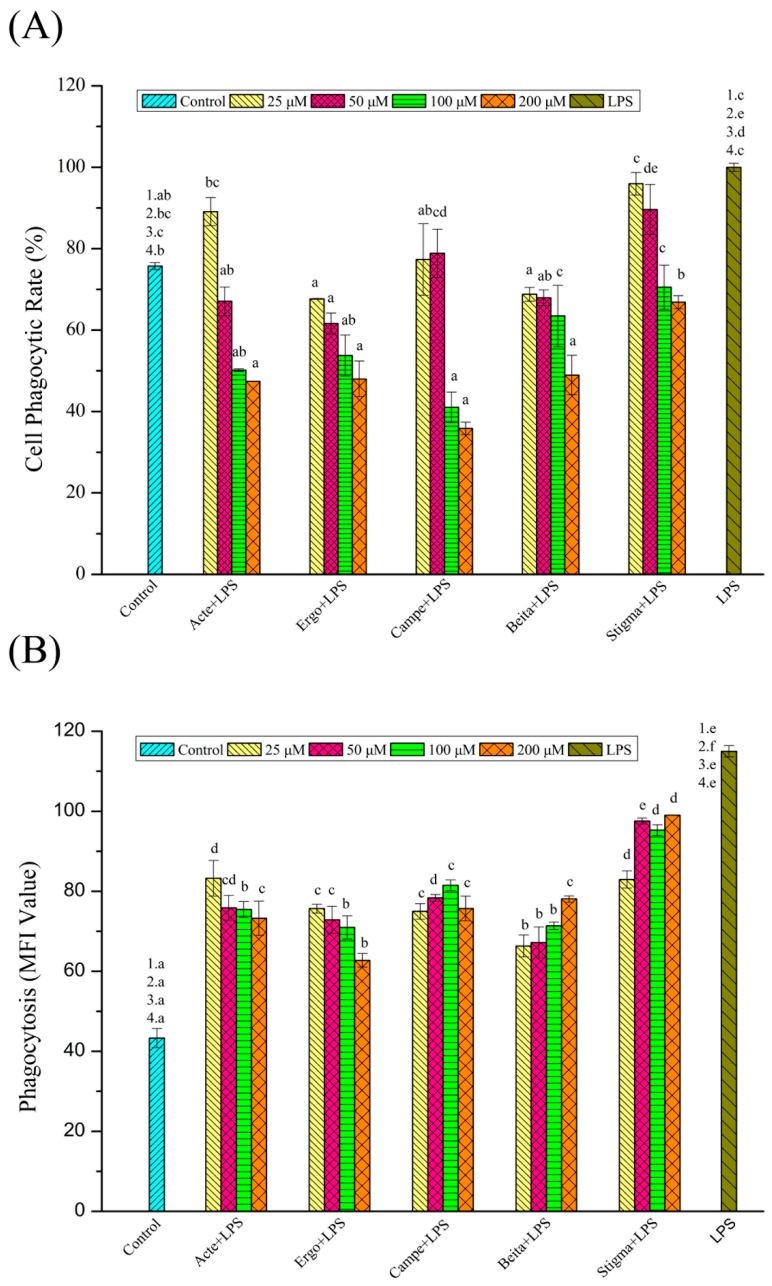
(**A**) Effects of phytosterols on the phagocytic effect of LPS-stimulated RAW264.7 macrophages detected by neutral red method (*n* = 6). (**B**) Effects of phytosterols on LPS-stimulated inflammatory cell model phagocytosis of fluorescein isothiocyanate (FITC)-dextran (*n* = 6). Acte (ergosterol acetate); Ergo (ergosterol); Campe (campesterol); Beita (β-sitosterol); Stigma (stigmasterol). The data are expressed as means ± SEM; different superscript letters for columns at the same concentration denote a significant difference (*p* < 0.05); the numbers describe that the control/model group was compared with the experimental group at the concentrations of 25 (1), 50 (2), 100 (3), and 200 (4) μM.

**Figure 4 foods-08-00582-f004:**
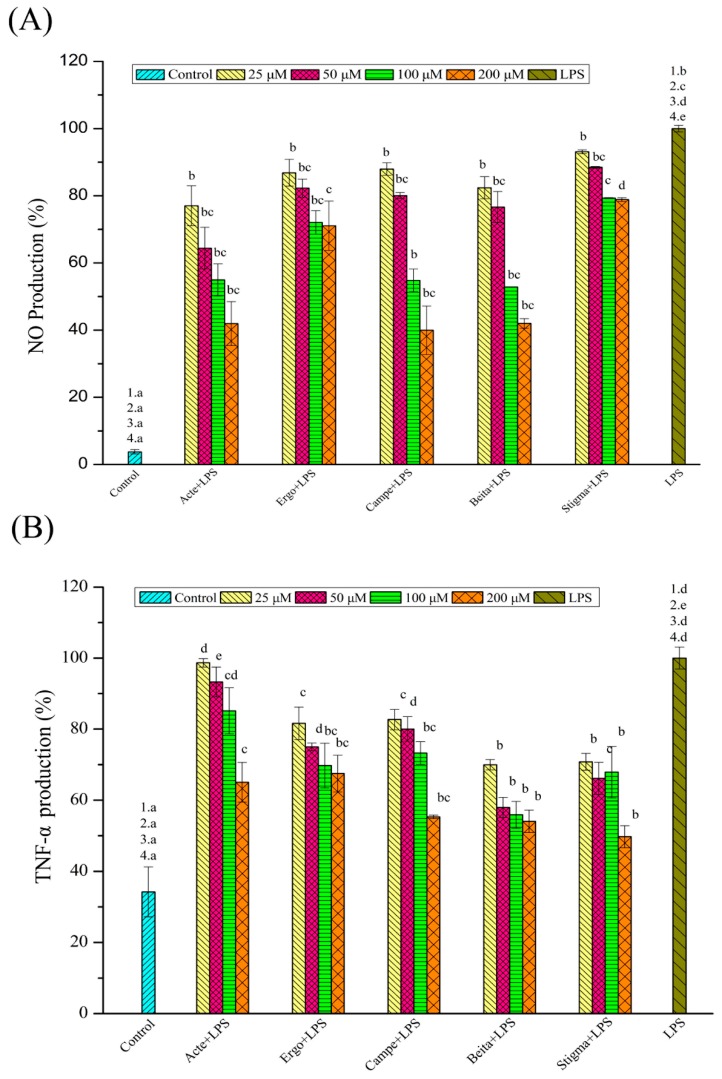
(**A**) Effects of phytosterols on the release of nitric oxide (NO) in LPS-stimulated inflammatory cell models (*n* = 6). Acte (ergosterol acetate). (**B**) Effects of phytosterol compounds on LPS-stimulated inflammatory cell model secreting cytokines tumor necrosis factor-α TNF-α (*n* = 6). Ergo (ergosterol); Campe (campesterol); Beita (β-sitosterol); Stigma (stigmasterol). The data are expressed as means ± SEM; different superscript letters for columns at the same concentration denote a significant difference (*p* < 0.05); the numbers describe that the control/model group was compared with the experimental group at the concentrations of 25 (1), 50 (2), 100 (3), and 200 (4) μM.

**Figure 5 foods-08-00582-f005:**
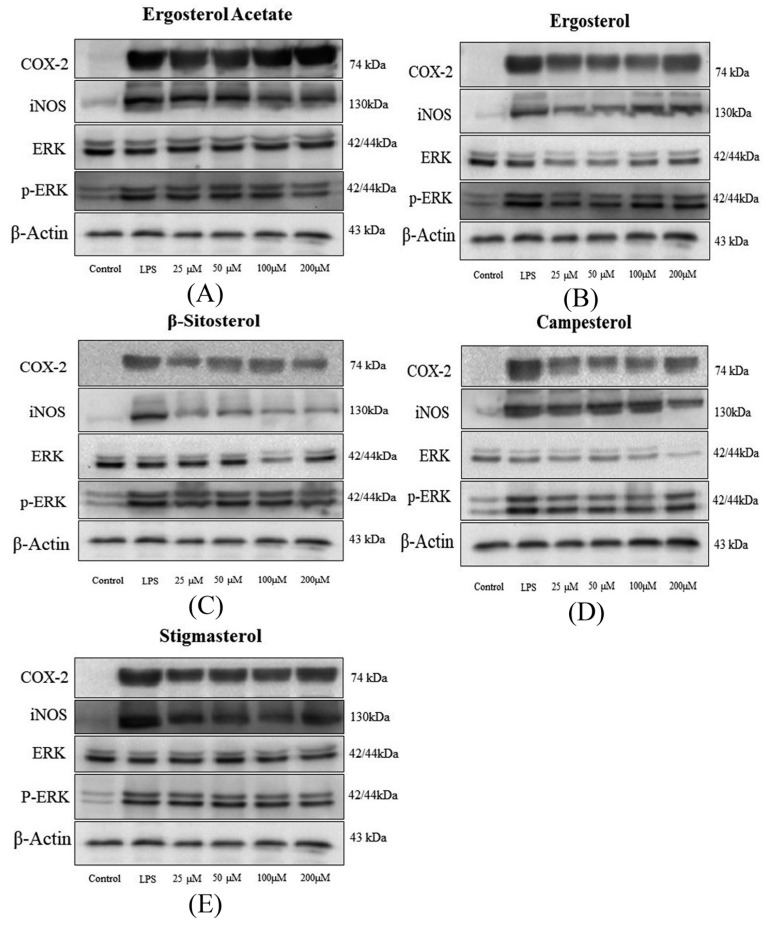
Results of Western blot for the expression of inflammatory regulators (cyclooxygenase-2 (COX-2), inducible nitric oxide synthase (iNOS), extracellular signal-regulated protein kinase (ERK), phosphorylated ERK (p-ERK), and β-actin): (**A**) ergosterol acetate; (**B**) ergosterol; (**C**) β-sitosterol; (**D**) campesterol; (**E**) stigmasterol.

**Figure 6 foods-08-00582-f006:**
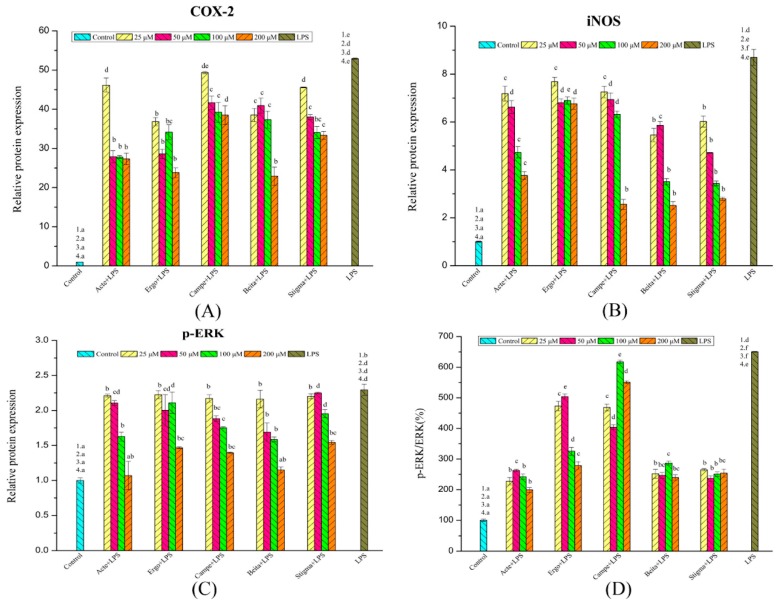
Effects of phytosterols on the expression of inflammatory regulators (**A**) COX-2, (**B**) iNOS, (**C**) p-ERK, and (**D**) p-ERK/ERK. Acte (ergosterol acetate); Ergo (ergosterol); Campe (campesterol); Beita (β-sitosterol); Stigma (stigmasterol). The data are expressed as means ± SEM; different superscript letters for columns at the same concentration denote a significant difference (*p* < 0.05); the numbers describe that the control/model group was compared with the experimental group at the concentrations of 25 (1), 50 (2), 100 (3), and 200 (4) μM.

**Figure 7 foods-08-00582-f007:**
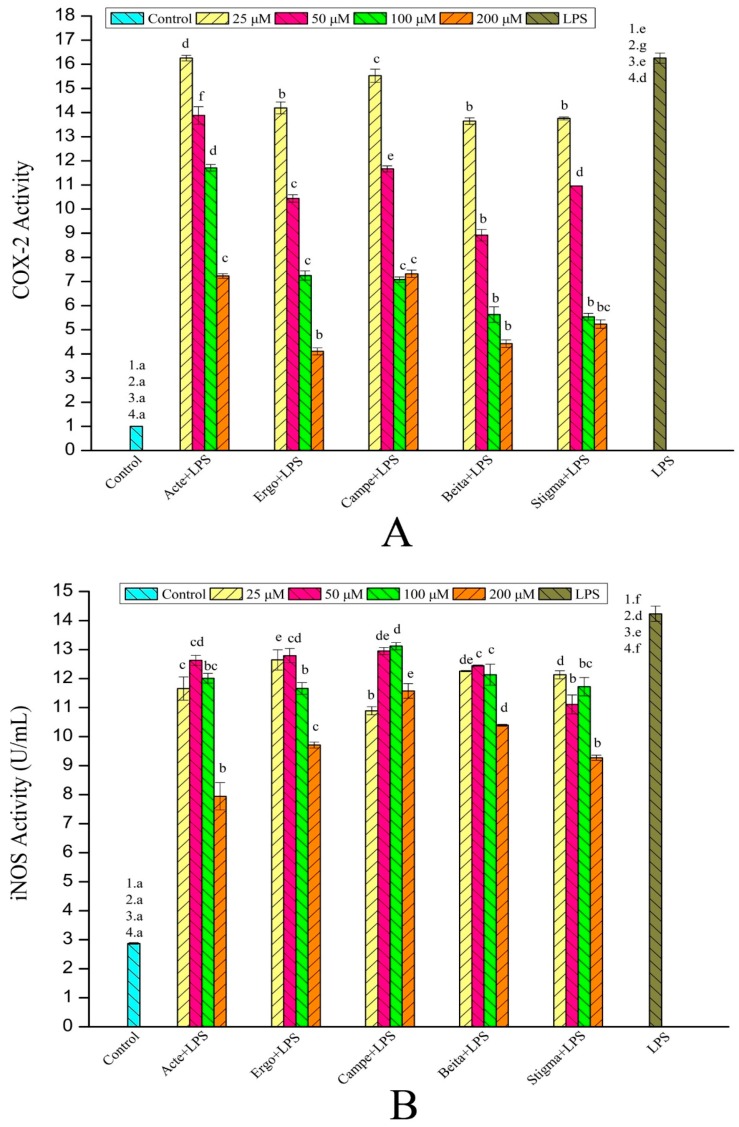
Effects of phytosterols on the activities of inflammatory regulators: (**A**) COX-2, (**B**) iNOS. Acte (ergosterol acetate); Ergo (ergosterol); Campe (campesterol); Beita (β-sitosterol); Stigma (stigmasterol). The data are expressed as means ± SEM; different superscript letters for columns at the same concentration denote a significant difference (*p* < 0.05); the numbers describe that the control/model group was compared with the experimental group at the concentrations of 25 (1), 50 (2., 100 (3), and 200 (4) μM, respectively.

**Table 1 foods-08-00582-t001:** Comparison of anti-inflammatory effects between phytosterols with different structure. FITC—fluorescein isothiocyanate; NO—nitric oxide; TNF-α—tumor necrosis factor-α; COX-2—cyclooxygenase-2; iNOS—inducible nitric oxide synthase; p-ERK—phosphorylated extracellular signal-regulated kinase.

Phytosterols	Phagocytosis (Neutral Red)	Phagocytosis (FITC-Dextran)	NO	TNF-α	COX-2 Expression	COX-2 Activity	iNOS Expression	iNOS Activity	p-ERK	p-ERK/ERK
Ergo vs. Acte ^a^	> ^b^	>	-- ^c^	>	>	>	< ^d^	<	<	<
Beita vs. Stigma	>	>	>	>	>	>	<	--	>	--
Beita vs. Campe	--	>	--	>	>	>	>	>	>	>

^a^ Ergo (ergosterol); Acte (ergosterol acetate); Beita (β-sitosterol); Stigma (stigmasterol); Campe (campesterol). ^b^ For this indicator, the former is significantly stronger than the latter in this comparison. ^c^ No significant differences were observed (*p* < 0.05). ^d^ For this indicator, the former is significantly weaker than the latter in this comparison.

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
