# Peer review of "Phytosterols Suppress Phagocytosis and Inhibit Inflammatory Mediators via ERK Pathway on LPS-Triggered Inflammatory Responses in RAW264.7 Macrophages and the Correlation with Their Structure"

_foods, 2019, doi:10.3390/foods8110582_

Round 1
Reviewer 1 Report
The paper is documented to show an anti-inflammatory effect of four selected phytosterols on a LPS-stimulated mouse macrophage cell line with structure-activity relationship of the phytosterols. The data clearly demonstrated that all of the phytosterols tested attenuated inflammatory responses induced by LPS. However, regarding the structure-activity relationship, no consistent tendency cannot be found between parameters, even between NO production and iNOS expression. Thus, I recommend that the relevant description in the Discussion section should be thoroughly rephrased. In addition, crucial mistakes and grammatical mistakes in English are found here and there. The revised manuscript should be checked by a native. Some of these mistakes are listed below.
Line 18
The anti-inflammatory activity of β-sitosterol were higher → “were” should be “was.
Line 42
Nowadays, although steroidal anti-inflammatory drugs such as aspirin, phenylbutazone ・・・
Aspirin and phenylbutazone are NSAIDS but not steroidal drugs.
Line 106
“Phosphate buffered solution” should be “Phosphate buffered saline”.
Line 116
“β-actin antibody” should be “anti-β-actin antibody”.
Line 144
Why the cell density was fully different from that in the CCK-8 assay?
Line 164
Why was the reaction terminated by the addition of 1 mL PBS?
Line 172
What does “supernatant cells” mean?
Line 178
No subject was found in “Treated as above”.
Line 201
How could you assess enzyme activity by Western blotting?
Line 219
Tukey-Kramer multiple comparison test should be applied to all the groups tested. Statistical analysis for each concentration is not desirable.
Line 334
the NO production induced by LPS in RAW264.7 macrophages were significantly →“were” should be “was.
Line 363
What does “TNF-αscavenging ability” mean? The phytosterols tested have an ability to scavenge or capture TNF-α?
Line 384
The inhibition of phytosterols on TNF-α production were → “were” should be “was.
Line 480
Should be “Fig. 6”.
Author Response
Response to Reviewer 1 Comments
Point 1: The paper is documented to show an anti-inflammatory effect of four selected phytosterols on a LPS-stimulated mouse macrophage cell line with structure-activity relationship of the phytosterols. The data clearly demonstrated that all of the phytosterols tested attenuated inflammatory responses induced by LPS. However, regarding the structure-activity relationship, no consistent tendency cannot be found between parameters, even between NO production and iNOS expression. Thus, I recommend that the relevant description in the Discussion section should be thoroughly rephrased.
Response 1: Thank you very much for your great support for acceptance of our manuscript in Foods. We feel very lucky that our manuscript went to you as the valuable comments from you helped us with the improvement of our manuscript. In the Discussion section (Line 527-531), I am sorry with the unclear statements in this section, we have changed “and supressed expression of iNOS is consistent with the decreased release of NO in LPS-stimulated macrophages” to “both NO production and iNOS expression were suppressed by phytosterols in LPS-stimulated macrophages”. And regarding the structure-activity relationship (Line 550-558), we further clarified the comparison among phytosterols and clearly demonstrated that the differences results on NO production and iNOS expression.
Point 2: In addition, crucial mistakes and grammatical mistakes in English are found here and there. The revised manuscript should be checked by a native. Some of these mistakes are listed below.
Response 2: Thank you very much. I apologize for our grammatical mistakes in English in the manuscript. We have already revised these mistakes in the revised manuscript.
Minor points;
Line 18
The anti-inflammatory activity of β-sitosterol were higher → “were” should be “was.
Response: “were” has been changed to “was”.
Line 42
Nowadays, although steroidal anti-inflammatory drugs such as aspirin, phenylbutazone ・・・
Aspirin and phenylbutazone are NSAIDS but not steroidal drugs.
Response: “Nowadays, although steroidal anti-inflammatory drugs such as aspirin, phenylbutazone …” → changed “steroidal” to “nonsteroidal”.
Line 106
“Phosphate buffered solution” should be “Phosphate buffered saline”.
Response: We have changed “Phosphate buffered solution” to “Phosphate buffered saline”.
Line 116
“β-actin antibody” should be “anti-β-actin antibody”.
Response: We have changed “β-actin antibody” to “anti-β-actin antibody”.
Line 144
Why the cell density was fully different from that in the CCK-8 assay?
Response: For better research, the cell density of CCK-8 assay was from pre-experiment and that of Neutral Red experiment was from the existing research (Such as: Wang, M., Zhao, S., Zhu, P., Nie, C., Ma, S., Wang, N., ... & Zhou, Y. (2018). Purification, characterization and immunomodulatory activity of water extractable polysaccharides from the swollen culms of Zizania latifolia. International journal of biological macromolecules, 107, 882-890).
Line 164
Why was the reaction terminated by the addition of 1 mL PBS?
Response: I am sorry to make this mistake, the “the reaction was terminated by adding 1 mL of PBS” was changed to “1 mL of PBS was added”.
Line 172
What does “supernatant cells” mean?
Response: I am sorry to make this mistake, we have changed “supernatant cells” to “cell culture supernatant”.
Line 178
No subject was found in “Treated as above”.
Response: I am sorry to make this mistake, it was treated as above (Section 2.5).
Line 201
How could you assess enzyme activity by Western blotting?
Response: I sorry am to make this mistake, we have changed “enzyme activity” to “proteins expression”.
Line 219
Tukey-Kramer multiple comparison test should be applied to all the groups tested. Statistical analysis for each concentration is not desirable.
Response: In order to better compare the effects of different phytosterols at the same concentration, we used “Duncan's multiple range test” for the analysis of each concentration. I am sorry with the deviation in this section, we have changed “Tukey-Kramer multiple comparison test” to “Duncan's multiple range test”.
Line 334
the NO production induced by LPS in RAW264.7 macrophages were significantly → “were” should be “was.
Response: “the NO production induced by LPS in RAW264.7 macrophages were significantly” → changed “were” to “was.
Line 363
What does “TNF-α scavenging ability” mean? The phytosterols tested have an ability to scavenge or capture TNF-α?
Response: I am sorry with the wrong statements in this section, we have changed “the TNF-α scavenging ability of phytosterols” to “the suppression of phytosterols on TNF-α production”.
Line 384
The inhibition of phytosterols on TNF-α production were → “were” should be “was.
Response: “The inhibition of phytosterols on TNF-α production were” → changed “were” to “was.
Line 480
Should be “Fig. 6”.
Response: We have changed “Fig. 5” to “Fig. 6”.

Reviewer 2 Report
The work in interesting but, in opinion of this referee, it has not been well designed and reported. Some major concerns are the following:
The anti-inflammatory effects of 5 phytosterols at doses of 25, 50, 100 and 200 μM are tested, but not including any point between the last two doses (100 and 200). This would be particularly necessary given the differences in the dose-response effect. Line 224. The authors talk about “control and model group” but they do not explain the differences between them. Most probably they mean untreated group and treated with LPS, which one is each one? The anti-inflammatory effects are sometimes directly proportional to the doses tested, but it doesn't always happen; and in fact sometimes the effect declines from a certain point. In the results and conclusions sections, the global effect is compared, without specifying what dose they refer to, drawing conclusions not consistent with what is represented in the figures. In that sense, the writing becomes chaotic and inconsistent, as the results mentioned by the authors do not fit with the results represented in the figures. The effect of phytosterols on the expression of anti-inflammatory enzymes (COX‐2, iNOS, p‐ERK and p‐ERK/ERK) is consistent, but the enzymatic activity should be also evaluated. Regarding basic background, some errors or statements are inadmissible. For instance: 30-31. Authors: ”However, chronic inflammation is harmful to the body, and it will induce a variety of diseases, such as arthritis, asthma, multiple sclerosis, inflammatory bowel disease and atherosclerosis”. It should be said that: “chronic inflammation is involved in pathologies such as ..... and aggravates the evolution of atherosclerosis” 42-43. Authors: “Nowadays, although steroidal anti‐inflammatory drugs such as aspirin, phenylbutazone have been used in the treatment of inflammation induced by tissue damage, they have side effects in clinical trials”. Aspirin and phenylbutazone are both NON steroidal anti‐inflammatory; and of course it is no needed a clinical trial to know about their side effects.Author Response
Response to Reviewer 2 Comments
Point 1: The work in interesting but, in opinion of this referee, it has not been well designed and reported. Some major concerns are the following:
The anti-inflammatory effects of 5 phytosterols at doses of 25, 50, 100 and 200 μM are tested, but not including any point between the last two doses (100 and 200). This would be particularly necessary given the differences in the dose-response effect. Line 224.
Response 1: Thank you very much for your great support for acceptance of our manuscript in Foods. The selection of experimental concentration is determined based on the pre-experiment and previous studies. However, your suggestion is reasonable. We will consider your suggestion in the follow-up study.
Point 2: The authors talk about “control and model group” but they do not explain the differences between them. Most probably they mean untreated group and treated with LPS, which one is each one?
Response 2: In Section 2.3 and 2.5, we have explained that model group: the cells were treated with culture medium containing 1 μg/mL LPS; control group: culture medium without phytosterols and LPS”.
Point 3: The anti-inflammatory effects are sometimes directly proportional to the doses tested, but it doesn't always happen; and in fact sometimes the effect declines from a certain point. In the results and conclusions sections, the global effect is compared, without specifying what dose they refer to, drawing conclusions not consistent with what is represented in the figures. In that sense, the writing becomes chaotic and inconsistent, as the results mentioned by the authors do not fit with the results represented in the figures.
Response 3: Thank you very much for your suggestions. We have rewritten this part, In the discussion and conclusion section, comparison between the structure-activity relationship of phytosterol's anti-inflammatory activity was based on the minimum effective concentration of phytosterols and the inhibitory effects of phytosterols at the same level, and the drawing conclusions were consistent with what is represented in the figures, we hope it can get your approval in this respect. Thank you.
Point 4: The effect of phytosterols on the expression of anti-inflammatory enzymes (COX‐2, iNOS, p‐ERK and p‐ERK/ERK) is consistent, but the enzymatic activity should be also evaluated.
Response 4: Thank you for your good suggestion. Your suggestion will be considered in our further studies.
Point 5: Regarding basic background, some errors or statements are inadmissible. For instance: 30-31. Authors: “However, chronic inflammation is harmful to the body, and it will induce a variety of diseases, such as arthritis, asthma, multiple sclerosis, inflammatory bowel disease and atherosclerosis”. It should be said that: “chronic inflammation is involved in pathologies such as ..... and aggravates the evolution of atherosclerosis”.
Response 5: According to your guideline, we have changed “However, chronic inflammation is harmful to the body, and it will induce a variety of diseases, such as arthritis, asthma, multiple sclerosis, inflammatory bowel disease and atherosclerosis” to “However, chronic inflammation is involved in pathologies such as arthritis, asthma, multiple sclerosis, inflammatory bowel disease and atherosclerosis, which are harmful to the body”.
Point 6: 42-43. Authors: “Nowadays, although steroidal anti‐inflammatory drugs such as aspirin, phenylbutazone have been used in the treatment of inflammation induced by tissue damage, they have side effects in clinical trials”. Aspirin and phenylbutazone are both NON steroidal anti‐inflammatory; and of course it is no needed a clinical trial to know about their side effects.
Response 6: I am sorry with the wrong statements in this section, we have changed “steroidal” to “nonsteroidal”, and “they have side effects in clinical trials” was from previous studies, thank you.

Round 2
Reviewer 1 Report
Please see the attachment.

Author Response
Response to Reviewer 1 Comments
Regarding the relation between NO production and ordiNOS expression, a discrepancy was found in the effect of stigmasterol as marked up by red circles below. Please make some explanation on this matter.
Response: Thank you very much for your great support for acceptance of our manuscript in Foods. In fact, for all phytosterols tested, their effects on NO production and iNOS expression were not consistent in our study At the low concentration (25 and 50 μM), no significant differences were observed for their effects on NO production, while the inhibition effects of β-sitosterol and stigmasterol on iNOS expression were significantly stronger than those of other phytosterols (p<0.05). And at higher concentrations (100 and 200 μM), phytosterols except stigmasterol exhibited similar effects on NO production and the stigmasterol group showed the lowest inhibiting activity; on the contrary, their inhibition effects on iNOS expression were in the order of β-sitosterol, stigmasterol > ergosterol acetate, campesterol > ergosterol. As shown in the following figure(as shown in Word file), NO production was regulated by complex factors in inflammatory response, it was reported that the inhibition of iNOS and cyclooxygenase-2 (COX-2) lead to the reduction of lipopolysaccharide (LPS)-induced NO production when an overproduction of NO was observed in LPS-stimulated models, in other word, the NO production was influenced by iNOS and COX-2 in macrophage [1-4]. To fully understand this, more experiments such as metabolomics should be conducted, which is the object of our further studies. Thank you very much.
[1] Chang, C. H., Wen, Z. H., Wang, S. K., Duh, C. Y. (2008). New anti-inflammatory steroids from the formosan soft coral Clavularia viridis. Steroids, 73, 562-567.
[2] De Tejada, G. M., Heinbocke, L., Ferrer-Espada, R., Heine, H., Alexander, C., et al. (2015). Lipoproteins/peptides are sesis-inducing toxins from bacteria that can be neutralized by synthetic antiendotoxin peptides. Scientific Reports, 5, 14292.
[3] Fernando, I. P., Sanjeewa, K. K., Kim, H. S., Kim, S. Y. (2017). Identification of sterols from the soft coral Dendronephthya gigantea and their anti-inflammatory potential. Environmental Toxicology and Pharmacology, 55, 37-43.
[4] Hu, J., Yang, B., Lin, X., Zhou, X., Yang, X., Long, L., Liu, Y. (2011). Chemical and biological studies of soft corals of the Nephtheidae family. Chemistry & Biodiversity, 8, 1011-1032.

Reviewer 2 Report
The writing of the work has improved, allowing to understand better results and conclusions. However, some conceptual mistakes and inaccuracies in the design and writing persist:
1. Figure 2. Effects of phytosterols on the proliferation. The concepts of cell proliferation and cell viability are confused, since evidently the viability of a culture cannot exceed 100%
2. The effect of phytosterols on the expression of anti-inflammatory enzymes (COX‐2, iNOS, p‐ERK and p‐ERK/ERK) is consistent. Taking in account that NSAID act as direct antagonist in most of the cases (like COX-2), to confirm the anti-inflammatory activity of these phytosterols, enzymatic activity should be evaluated.
3. In point 6, it would be a better option to write "they have been shown to have harmful side effects in clinical practice".
4. Line 125: The anti‐β‐actin antibody; it is enough to put β‐actin antibody
Author Response
Response to Reviewer 2 Comments
Point 1: The writing of the work has improved, allowing to understand better results and conclusions. However, some conceptual mistakes and inaccuracies in the design and writing persist:
Figure 2. Effects of phytosterols on the proliferation. The concepts of cell proliferation and cell viability are confused, since evidently the viability of a culture cannot exceed 100%.
Response 1: Thank you very much for your great support for acceptance of our manuscript in Foods. You are right, I am sorry to make this mistake, the “cell viability” were changed to “cell proliferation”.
Point 2: The effect of phytosterols on the expression of anti-inflammatory enzymes (COX‐2, iNOS, p‐ERK and p‐ERK/ERK) is consistent. Taking in account that NSAID act as direct antagonist in most of the cases (like COX-2), to confirm the anti-inflammatory activity of these phytosterols, enzymatic activity should be evaluated.
Response 2: Thank you very much for your suggestion, enzymatic activity should be evaluated to confirm the anti-inflammatory activity of tested phytosterols, thus we did supplementary experiment to evaluate the COX-2 and iNOS activity.In most cases, the results were consistent with those of COX-2/iNOS expression (Section 3.6 and Figure 7). Thank you.
Point 3: In point 6, it would be a better option to write "they have been shown to have harmful side effects in clinical practice".
Response 3: According to your guideline, we have changed “they have side effects in clinical trials” to “they have been shown to have harmful side effects in clinical practice”.
Point 4: Line 125: The anti‐β‐actin antibody; it is enough to put β‐actin antibody
Response 4: Thank you for your good suggestion. We have changed “anti‐β‐actin antibody” to “β‐actin antibody”.
